Usefulness of the cytokines expression of Th1/Th2/Th17 and urinary CD80 excretion in adult-onset minimal change disease

Chen Ping 1 2
Chen Yan 3
Jiang Maoqing 4
Mo Yijun 5
Ying Huanhuan 5
Tang Xun 1
Zhang Jun gzh_zj@qq.com 1
1 Department of Nephrology, Zhujiang Hospital of Southern Medical University , Guangzhou , Guangdong , China
2 Department of Nephrology, Ningbo First Hospital , Ningbo , Zhejiang , China
3 Department of Physical Examination, Ningbo First hospital , Ningbo , Zhejiang , China
4 Department of Nuclear Medicine, Ningbo First Hospital , Ningbo , Zhejiang , China
5 Department of Laboratory Medicine, Ningbo First Hospital , Ningbo , Zhejiang , China
Albertini Maria Cristina
Electronic publication date: 2020 Sep 8
Publication date: 2020
Volume: 8
Electronic Location ID: e9854
Received 2020 May 14; Accepted 2020 Aug 11
Copyright: ©2020 Chen et al.
Copyright year: 2020
Copyright holder: Chen et al.
License: This is an open access article distributed under the terms of the Creative Commons Attribution License, which permits unrestricted use, distribution, reproduction and adaptation in any medium and for any purpose provided that it is properly attributed. For attribution, the original author(s), title, publication source (PeerJ) and either DOI or URL of the article must be cited.
License URL: https://creativecommons.org/licenses/by/4.0/

Keywords: Th1/Th2/Th17, CD80, Adult-onset minimal change disease

Funding: Medical and Health Technology Program in Zhejiang province No. 2018KY679 Ningbo medical science and technology program 2019Y08 This work was supported by the project of Medical and Health Technology Program in Zhejiang province (No. 2018KY679) and Ningbo medical science and technology program (No. 2019Y08). The funders had no role in study design, data collection and analysis, decision to publish, or preparation of the manuscript.

==============================
Background

Minimal change disease (MCD) is a common form of nephrotic syndrome in adults. However, the molecular mechanism underlying the pathogenesis of MCD remains incompletely understood. In this study, we aimed to investigate the role of the cytokines expression of Th1/Th2/Th17 and urinary CD80 excretion in adult-onset MCD patients.

Methods

The lymphocyte subsets, 34 cytokine levels of Th1/Th2/Th17, serum and urine concentrations of CD80, and expression of CD80 in glomeruli were analyzed in 28 cases (15 males and 13 females; average age: 34.1 years, age range: 18–56 years), including 10 patients with MCD in relapse, nine patients with MCD in remission and nine healthy controls.

Results

There was no significant difference of CD3+CD4+ cells proportion among patients with MCD in relapse, MCD in remission and healthy controls (P = 0.802). The cytokine levels of GM-CSF and tumor necrosis factor (TNF)-related activation-induced cytokine (TRANCE) in patients with MCD in relapse increased 1.5 times higher than those in remission. An evident increase in the excretion of urinary CD80 was found in patients with relapsed MCD compared with those in remission (598.4 ± 115.8 vs 81.78 ± 7.04 ng/g creatinine, P < 0.001) and healthy controls (598.4 ± 115.8 vs 67.44 ±  8.94  ng/g creatinine, P < 0.001). CD80 expression was observed in podocyte of MCD patient in relapse by immunofluorescence technique.

Conclusions

The cytokines GM-CSF and TRANCE are increased and the urinary CD80 levels are elevated in adult-onset MCD patients in relapse, indicating a disorder of Th1/Th2/Th17 balance and that the elevated excretion of CD80 may underlie the pathogenesis and development of adult-onset MCD.

Introduction

Minimal change disease (MCD) is commonly regarded as due to T cell subsets disorder and certain circulating cytokines that trigger dysfunction of podocytes and resulting in proteinuria (Boumediene et al., 2018; Kaneko et al., 2015; Iwabuchi et al., 2018; Wang & Greenbaum, 2019). However, not much is known about which type of T cell subset and cytokine play a critical role in the pathogenesis of MCD. Previous studies revealed that patients with MCD were characterized by downregulation of Th1 cytokines and predominance of Th2 and Th17, which might be harmful factors of glomerulus, leading to the occurrence and development of MCD (Le Berre et al., 2005; Salcido-Ochoa et al., 2017). However, Kaneko et al. (2002) reported that the percentages of Th1, Th2 and the ratio of Th1/Th2 showed no significant differences between nephrotic relapse, remission and healthy controls in childhood. Thus, the cytokines expression of Th1/Th2/Th17 still needs to be characterized and evaluated in MCD patients.

CD80, also known as B7-1, is a costimulatory molecule of T cells and is also involved in T cell activation and termination (Novelli, Benigni & Remuzzi, 2018). Reiser et al. (2004) demonstrated that induction of CD80 expression on podocyte led to reorganization of actin cytoskeleton that modified glomerular permselectivity and caused proteinuria in mice. Moreover, it has been proposed that MCD is a “two-hit” podocyte immune disorder (Shimada et al., 2011). The first hit is the induction of CD80 in podocyte by various stimuli and the second hit is the ineffective censoring of podocyte CD80 due to a defective autoregulation by podocyte itself. Besides, urinary CD80 is elevated in MCD in children and measuring urinary CD80 concentrations could distinguish MCD from focal segmental glomerulosclerosis (FSGS) (Ling et al., 2015; Garin et al., 2010; Garin et al., 2009). Recently, a CD80 inhibitor, abatacept (cytotoxic T-lymphocyte-associated antigen 4-immunoglobulin fusion protein, CTLA-4-Ig), was used as a therapeutic drug in “CD80-positive kidney disease” in MCD patients even though in only a limited number of cases (Garin et al., 2015). However, few studies have tried to explain the possible reasons why elevated urinary levels of CD80 excretion are encountered in adult-onset MCD.

Therefore, in this study, we aimed to characterize the serum cytokines expression of Th1/Th2/Th17 and urinary CD80 excretion in adult-onset patients with MCD and healthy controls, as well as their role in the pathogenesis and development of MCD.

Materials & Methods

Patient selection

Adult patients with biopsy proven MCD and healthy volunteers were studied (Table 1). None of the healthy control subjects involved in the study had any underlying immunologic disease in the study. Patients with renal dysfunction were excluded (glomerular filtration rate<60 ml/min or serum creatinine>1.5 mg/dl). MCD was defined according to the established pathology criteria (Vivarelli et al., 2017). Relapse of MCD was defined as proteinuria (≥3+ using the tetrabromophenol-citrate buffer colorimetric qualitative dipstick test or urinary protein/creatinine ratio >3.0 mg/mg), edema and hypoalbuminemia (of<30 g/l). Remission of MCD was defined as a urinary protein/creatinine ratio<0.3 mg/mg or no proteinuria using the colorimetric qualitative test. Our study was approved by the Medical Ethics Committee of Ningbo First Hospital (No. 2017-R033). A written informed consent was obtained from all participants.

Table 1 Characteristic of patients with MCD in relapse, MCD in remission and healthy controls.

No. of patients	Age	Gender	Serum Alb (g/l)	eGFR (ml/ min/1.73 m2)	Up/Uc ratio	Urinary CD80 (ng/g Cre)	Urinary CTLA-4 (ng/g Cre)	Treatment	
MCD in relapse	
1	44	M	18.7	117.4	10.5	271.00	158.00	None	
2	30	M	29	108.5	3.7	737.00	209.00	Pre10 mg/d	
3	28	M	28	99.9	7	199.00	147.00	Pre 5 mg/d	
4	26	F	24	131.6	15	1050.00	200.00	Pre 20 mg QD	
5	46	M	26.5	122.9	3.5	864.00	157.00	Met 12 mg QD, Tac 1 mg BID	
6	18	F	12.5	97.7	22.6	503.00	178.00	Pre 15 mg every other day	
7	50	F	25.5	110.8	11	560.00	196.00	Met 8 mg QD	
8	35	M	24.7	120.5	8.5	1250.00	150.00	Pre 5 mg/d	
9	22	F	19.8	100.1	25.9	240.00	207.00	None	
10	40	M	23.4	130.8	18	310.00	135.00	Pre 10 mg every other day	
mean ± SEM	33.9 ± 3.42		23.21 ± 1.57	114 ± 3.98	12.48 ± 2.47	598.4 ± 115.8	173.7 ± 8.73		
MCD in remission	
11	28	M	35	125.9	0.12	90.00	142.00	Pre 30 mg/d	
12	30	M	40	138	Neg	79.00	139.00	Pre 20 mg/d	
13	45	F	32	125.6	0.18	60.00	127.00	Met 16 mg QD, Tac 1 mg BID	
14	22	M	34	115	0.11	110.00	189.00	Pre 60 mg/d	
15	35	F	39	118	0.06	120.00	199.00	Met 12 mg/d	
16	19	F	40	140	0.05	54.00	165.00	Met 12 mg every other day	
17	51	M	45	130	0.01	68.00	155.00	Pre 20 mg/d	
18	33	F	47	99	Neg	73.00	153.00	Pre15 g/d	
19	29	M	55	111	Neg	82.00	126.00	Pre 10 mg/d	
mean ± SEM	32.44 ± 3.408	40.78 ± 2.4	122.5 ± 4.386	0.05 ± 0.002	81.78 ± 7.038	155 ± 8.54		
Healthy controls	
20	30	F	45	120	Neg	100.00	200.00	N/A	
21	22	F	55	119	Neg	44.00	210.00	N/A	
22	56	F	50	110	Neg	32.00	168.00	N/A	
23	18	M	43	130	Neg	57.00	189.00	N/A	
24	26	F	45	124	0.04	69.00	146.00	N/A	
25	43	M	53	120	0.02	75.00	139.00	N/A	
26	34	M	52	118	0.01	110.00	132.00	N/A	
27	50	F	46	110	Neg	80.00	129.00	N/A	
28	44	M	44	112	Neg	40.00	127.00	N/A	
mean ± SEM	35.89 ± 4.36		48.11 ± 1.47	118.1 ± 2.214	0.007 ± 0.004	67.44 ± 8.94	160 ± 10.85		
Notes.

Abbreviation MCD minimal change disease

eGFR estimated glomerular filtration rate

Up/Uc urinary protein/urinary creatinine

M male

F female

Pre Prednisone

Met Methylprednisolone

Tac Tacrolimus

Cre Creatinine

Alb Albumin

NA not available

Clinical samples collection and cytokine antibody array measure

We randomly selected six patients with MCD in relapse, three patients with MCD in remission and three healthy controls. From January 2017 to January 2018, 10ml peripheral blood samples were collected from patients with MCD and healthy controls in Ningbo First Hospital, Medical College of Ningbo University. The total 34 cytokines of Th1/Th2/Th17 in MCD patients and healthy controls were detected by cytokine antibody array using a RayBio® human cytokine antibody array (RayBiotech, Inc, Norcross, GA, USA, and AAH-TH17-G1). Membranes were incubated with diluted antibodies at room temperature for 2 h. The detections were accomplished according to the manufacturer’s manual and previous research (Huang, 2001).

Lymphocyte subsets assessment

We used flow cytometry analysis to detect lymphocyte subsets in whole peripheral blood in ten patients with MCD in relapse, nine patients with MCD in remission and nine healthy controls. Briefly, peripheral blood was incubated with marked monoclonal antibodies for 20 min at room temperature in the dark. Then lysing reagent was added and incubated for 15 min at room temperature in the dark. The lymphocyte subsets analysis was performed by using a BD flow cytometer (BD Bioscience).

CD80 and CTLA-4 measurements

A commercially available ELISA kit (Bender Med-Systems, Burlingame, CA, USA) was used for measuring CD80 in blood and urine. We detected CTLA-4 in blood and urine according to previous study (Oaks & Hallett, 2000). We adjusted the results of CD80 and CTLA-4 with urinary creatinine excretion. Urinary creatinine and protein and serum albumin were detected by an auto-analyzer.

Immunohistochemistry

We used immunofluorescence technique to test the expression of CD80 in glomeruli of MCD patient in relapse. Snap-frozen renal specimens were incubated with monoclonal synaptopodin or WT-1 antibody (1:50; Santa Cruz, CA) for 2 h at room temperature to reveal podocyte. Then, we washed the sections three times with PBS and incubated specimens with anti-CD80 antibody (1:100) (goat; R&D Systems, Minneapolis, MN) 2 h at room temperature. After washing three times with PBS, we incubated sections with anti-goat 488 and chicken anti-mouse 594 Alexa Fluor antibodies (1:1500, Invitrogen, Carlsbad, CA) for 1 h at room temperature.

Statistical analysis

Graph Pad Instat version 5.0 was used to perform statistical analysis and data graphics. Non-parametric ANOVA (Kruskal-Wallis test) was conducted for statistical analysis. We used Mann–Whitney U test or Wilcoxon signed rank test (when applicable) to determine the differences between means. Spearman correlation coefficient was used to calculate the correlation between urinary CD80 and proteinuria. Values were expressed as means ± standard error of mean (SEM). Results were considered statistically significant if P < 0.05.

Results

Clinical characteristics

The clinical characteristics and laboratory results of patients with MCD in relapse (n = 10), MCD in remission (n = 9) and healthy controls (n = 9) are summarized in Table 1. A total of 28 cases were reported, 15 males and 13 females, with an average age of 34.1 ± 2.1 years (age range: 18–56 years). Patients with MCD in relapse were analyzed at onset of illness. Eight of the relapsed patients were on a tapering dose of immunosuppressive treatment while the remaining two were after drug withdrawal. All patients with MCD in remission were getting immunosuppressive treatment at the time of measuring.

Lymphocyte subsets in the peripheral blood of MCD patients and healthy controls

We analyzed lymphocyte subsets, including CD3+T cells, CD3+ CD4+ T cells, CD3+CD8+T cells, CD3−CD19+B cells and CD3−CD16+/CD56+ NK cells. The distribution of lymphocyte subsets in the peripheral blood of MCD patients and healthy controls are showed in Table 2. There were no statistically significant differences in the proportions of CD3+ and CD3+CD4+ cells among patients with MCD in relapse, MCD in remission and healthy controls (P = 0.445 and P = 0.802). MCD patients in relapse had significant higher proportions of CD3+CD8+ cells compared to MCD in remission (42.7 ± 2.29% vs. 27.42 ± 1.51%, P = 0.008) and healthy controls (42.7 ± 2.29% vs. 27.97 ± 2.34%, P<0.001). Furthermore, CD4+/CD8+ ratio was lower in MCD in relapse compared with MCD in remission (0.84 ± 0.09 vs. 1.45 ± 0.14, P = 0.014) and healthy controls (0.84 ± 0.09 vs. 1.4 ± 0.12, P = 0.005). The CD3−CD16+/CD56+ NK cell and CD3−CD19+B cell populations revealed no significant differences among patients with MCD in relapse, MCD in remission and healthy controls (P = 0.199 and P = 0.445).

Cytokine profiles in serum of MCD patients and controls

We investigated Th1/Th2/Th17 cytokines in the serum of the randomly selected MCD patients and controls by cytokine antibody array (Figs. 1A–1C and Table 3). The characteristics of the randomly selected MCD patients and healthy controls are presented in Table 4. Each cytokine antibody array included 34 cytokines (Fig. 1D). The concentrations that increased by ≥1.5-fold or decreased by ≤0.65-fold were considered as significant. Our results in Table 3 indicate that the levels of GM-CSF and tumor necrosis factor (TNF)-related activation-induced cytokine (TRANCE) in patients with MCD in relapse increased 1.5 times higher than in patients in remission. GM-CSF, IL-10, IL-22 and TNF beta levels increased significantly in patients with MCD in relapse compared to healthy controls. The expression of CD40 was found to have decreased in MCD in relapse compared to healthy controls.

Table 2 Lymphocyte subsets in patients with MCD in relapse, MCD in remission and healthy controls.

Variables	MCD in relapse
(n = 9)	MCD in remission
(n = 10)	Healthy controls
(n = 9)	
CD3+%	75.74 ± 2.36	72.59 ± 2.15	70.98 ± 2.85	
CD3+CD4+%	35.07 ± 2.02	37.45 ± 1.94	35.23 ± 2.4	
CD3+CD8+%	42.7 ± 2.29*#	27.42 ± 1.51	27.97 ± 2.34	
CD4+/CD8+	0.84 ± 0.09#*	1.45 ± 0.14	1.4 ± 0.12	
CD3−CD16+/ CD56+ NK%	11.44 ± 1.54	14.44 ± 1.4	11.52 ± 1.7	
CD3−CD19+B cells%	10.85 ± 1.14	15.10 ± 2.56	12.05 ± 1.12	
Notes.

* p < 0.05 compared MCD in relapse with healthy controls.

# P < 0.05 compared MCD in relapse with MCD in remission.

Figure 1 Cytokine profiles of patients with MCD and normal control.

Cytokine Ab array images from a patient with MCD in relapse (A), a patient in remission (B), and a healthy control (C) were shown. The levels of cytokine in serum were represented by the spot density. The four spots in the upper left and two in the lower right corners of the membranes indicate positive controls. The position of 34 human Th1-, Th2- and Th17–related cytokines in the antibody based microarray (D).

Table 3 The results of 34 human Th1-, Th2- and Th17-related cytokines in patients with MCD in relapse, MCD in remission and healthy controls (median).

Variables	MCD in relapse	MCD in remission	healthy controls	
CD30	0.05 (0.02–0.07)	0.07 (0.04–0.07)	0.07 (0.03–0.29)	
CD40 Ligand	0.43 (0.3–0.62)	0.42 (0.29–0.53)	0.51 (0.35–0.74)	
CD40	0.14 (0.07-0.24)#	0.20 (0.06–0.34)	0.21 (0.08–0.30)	
GCSF	0.04 (0.02–0.09)	0.05 (0.02-0.09)**	0.03 (0.01–0.05)	
GITR (TNFRSF18)	0.61 (0.44–0.75)	0.54 (0.40–0.61)	0.57 (0.47–0.64)	
GM-CSF	0.10 (0.02-0.26)*#	0.05 (0.02–0.06)	0.06 (0.06–0.1)	
IFN-gamma	0.54 (0.44–0.65)	0.52 (0.49–0.56)	0.49 (0.41–0.57)	
IL-1 R1	0.34 (0.20–0.58)	0.37 (0.34-0.44)**	0.23 (0.13–0.32)	
IL-1 R2	0.71 (0.62–0.78)	0.77 (0.73–0.82)	0.71 (0.61-0.81)	
IL-10	0.18 (0.05-0.33)#	0.16 (0.11–0.20)	0.11 (0.10–0.12)	
IL-12 p40	0.68 (0.60–0.76)	0.62 (0.61–0.64)	0.70 (0.61–0.64)	
IL-12 p70	0.46 (0.33–0.57)	0.38 (0.25–0.47)	0.49 (0.42–0.62)	
IL-13	0.20 (0.13–0.28)	0.16 (0.06–0.23)	0.20 (0.13–0.29)	
IL-17A	0.16 (0.10–0.25)	0.16 (0.07–0.21)	0.18 (0.14–0.26)	
IL-17F	0.16 (0.10–0.23)	0.16 (0.08–0.23)	0.23 (0.14–0.30)	
IL-17 RA	0.03 (0.01–0.05)	0.04 (0.03–0.06)	0.03 (0.03–0.06)	
IL-1 beta (IL-1 F2)	0.44 (0.30–0.62)	0.38 (0.27–0.45)	0.45 (0.26–0.65)	
IL-2	0.58 (0.45–0.68)	0.54 (0.47–0.58)	0.59 (0.46–0.65)	
IL-21	0.54 (0.41–0.66)	0.52 (0.48–0.57)	0.50 (0.40–0.58)	
IL-21 R	0.49 (0.35–0.67)	0.49 (0.44–0.54)	0.41 (0.34–0.50)	
IL-22	0.29 (0.07–0.39)#	0.27 (0.20–0.37)**	0.11 (0.10–0.12)	
IL-23	0.56 (0.44–0.71)	0.65 (0.60–0.68)	0.55 (0.47–0.62)	
IL-28A	0.22 (0.17–0.28)	0.20 (0.17–0.22)	0.25 (0.20–0.28)	
IL-4	0.21 (0.13–0.30)	0.16 (0.08–0.22)	0.23 (0.08–0.22)	
IL-5	0.07 (0.05–0.13)	0.05 (0.03-0.09)**	0.09 (0.07–0.12)	
IL-6	0.08 (0.04–0.11)	0.07 (0.05–0.11)	0.08 (0.06–0.12)	
IL-6 R	0.95 (0.89–0.98)	0.94 (0.88–0.98)	0.96 (0.93–0.99)	
MIP-3 α	0.35 (0.24–0.45)	0.32 (0.19–0.41)	0.30 (0.20–0.44)	
gp130	0.86 (0.82–0.90)	0.87 (0.83–0.90)	0.88 (0.85–0.92)	
TGF beta 1	0.45 (0.34–0.58)	0.46 (0.38–0.51)	0.48 (0.43–0.51)	
TGF beta 3	0.52 (0.40–0.68)	0.51 (0.48–0.56)	0.52 (0.50–0.57)	
TNF alpha	0.37 (0.24–0.65)	0.32 (0.30–0.34)	0.29 (0.23–0.40)	
TNF beta	0.45 (0.30–0.64)#	0.33 (0.29–0.35)	0.29 (0.24–0.37)	
TRANCE (TNFSF11)	0.22 (0.13–0.36)*	0.15 (0.07–0.20)	0.18 (0.12–0.21)	
Notes.

* P < 0.05 compared relapse with remission.

# P < 0.05 compared relapse with healthy controls.

** P < 0.05 compared remission with healthy controls.

Table 4 Characteristic of randomly selected patients with MCD in relapse, MCD in remission and healthy controls.

Patient	Age	Gender	Serum Alb (g/l)	eGFR (ml/ min/1.73 m2)	Up/Uc ratio	Urinary CD80 (ng/g Cre)	Urinary CTLA-4 (ng/g Cre)	Treatment	
MCD in relapse	
2	30	M	29	108.5	3.7	737.00	209.00	Pre10 mg/d	
4	26	F	24	131.6	15	1,050.00	200.00	Pre 20 mg QD	
5	46	M	26.5	122.9	3.5	864.00	157.00	Met 12 mg QD, Tac 1 mg BID	
6	18	F	12.5	97.7	22.6	503.00	178.00	Pre 15 mg every other day	
7	50	F	25.5	110.8	11	560.00	196.00	Met 8 mg QD	
10	40	M	23.4	130.8	18	310.00	135.00	Pre 10 mg every other day	
Mean ± SEM	35 ± 5.05		23.48 ± 2.34	117 ± 5.54	12.3 ± 3.15	670.7 ± 109	179.2 ± 11.6		
MCD in remission	
14	22	M	34	115	0.11	110.00	189.00	Pre 60 mg/d	
15	35	F	39	118	0.06	120.00	199.00	Met 12 mg/d	
19	29	M	55	111	Neg	82.00	126.00	Pre 10 mg/d	
mean ± SEM	28.67 ± 3.75	42.67 ± 6.33	114.7 ± 2.02	0.06 ± 0.03	104 ± 11.37	171.3 ± 22.85		
healthy controls	
22	56	F	50	110	Neg	32.00	168.00	N/A	
26	34	M	52	118	0.01	110.00	132.00	N/A	
28	44	M	44	112	Neg	40.00	127.00	N/A	
mean ± SEM	44 ± 6.36		48.67 ± 2.4	113.3 ± 2.4	0.003 ± 0.003	60.67 ± 24.77	142.3 ± 12.91		
Notes.

Abbreviation MCD minimal change disease

eGFR estimated glomerular filtration rate

Up/Uc urinary protein/urinary creatinine

M male

F female

Pre Prednisone

Met Methylprednisolone

Tac Tacrolimus

Cre Creatinine

Alb Albumin

NA not available

Measurement of CD80 and CTLA-4 expression

We explored urinary CD80 expression adjusted by urinary creatinine in patients with MCD in relapse (n = 10), MCD in remission (n = 9) and healthy controls (n = 9) (Fig. 2A). A significant increase in the excretion of urinary CD80 was observed in MCD patients in relapse when compared with patients in remission (598.4 ± 115.8 ng/g vs. 81.78 ± 7.04 ng/g creatinine, P<0.001) and healthy controls (598.4 ± 115.8 ng/g vs. 67.44 ± 8.94 ng/g creatinine, P <0.001). The excretion of urinary CD80 showed no significant difference between MCD patients in remission and healthy controls (P = 0.269, Fig. 2A). There was no correlation between urinary CD80 and proteinuria in MCD patients in relapse (r =  − 0.32, P = 0.366, Fig. 2B).

Figure 2 CD80 urinary concentrations (ng/g creatinine) in patients with MCD and healthy controls.

(A) Comparisons: P < 0.001 MCD in relapse vs. MCD in remission; P < 0.001 MCD in relapse vs. healthy controls. (B) Correlation between urinary CD80 and proteinuria in patients with MCD in relapse. (C) Urinary excretion of CTLA-4 (ng/g creatinine) in patients with MCD and control subjects. (D) Correlation between urinary CTLA-4 and proteinuria in MCD patients in relapse. (E) Correlation between urinary CD80 and urinary CTLA-4 in MCD patients in relapse. (F). Correlation between urinary CD80 and urinary CTLA-4 in MCD patients in remission.

In contrast to the urinary results, no significant differences were found in serum CD80 concentrations among patients with MCD in relapse, MCD in remission, and healthy control subjects (379.9 ± 32.3 vs. 287.6 ± 19.48 vs. 342.4 ± 28.43 pg/ml, P = 0.081). Urinary CTLA-4 levels were not significantly increased in MCD patients in relapse when compared with patients in remission (173.7 ± 8.73 vs. 155.0 ± 8.54 ng/g creatinine, P = 0.098) and healthy control subjects (173.7 ± 8.73 vs. 160.0 ± 10.85 ng/g creatinine, P = 0.253, Fig. 2C). There was no correlation between the concentrations of urinary CTLA-4 and proteinuria in MCD patients in relapse (r = 0.18, P = 0.632, Fig. 2D). There was also no inverse correlation between urinary CTLA-4 and urinary CD80 in MCD patients in relapse (r = 0.15, P = 0.682, Fig. 2E) nor in remission (r = 0.3, P = 0.437, Fig. 2F).

Compared with MCD patients in remission, the urinary CD80 to CTLA-4 ratio was elevated significantly in MCD in relapse (P = 0.004, Fig. 3A). No significant differences of serum CD80/CTLA-4 ratio (P = 0.278, Fig. 3B) and CTLA-4 concentrations (P = 0.949, Fig. 3C) were found among patients with MCD in relapse, MCD in remission and healthy controls.

Figure 3 CD80 and CTLA-4 in patients with MCD in relapse, MCD in remission and healthy controls.

(A) The ratio of CD80 (ng/g creatinine) to CTLA-4 (ng/g creatinine) in urine in patients with MCD in relapse, MCD in remission and healthy controls. (B) Serum CD80/CTLA-4 ratio between patients with MCD in relapse, MCD in remission and healthy controls. (C) Serum CTLA-4 concentrations in patients with MCD in relapse, MCD in remission and healthy controls.

The expression of CD80 in podocyte in patient with relapsed MCD

We used immunofluorescence technique to test the expression of CD80 in glomeruli of patient 5 as listed in Table 1. Two glomeruli from patient 5 with relapsed MCD were stained for CD80 in green (Figs. 4A and 4D), synaptopodin in red (Fig. 4B) and WT-1 in red (Fig. 4E). The double immunostaining for CD80 and synaptopodin in the glomerulus of MCD patient in relapse showed colocalization (Fig. 4C). CD80 and WT-1 co-localized in the glomerulus of MCD patient in relapse (Fig. 4F).

Figure 4 The expression of CD80 in glomeruli of MCD patient in relapse.

CD80 is expressed (green stain) in the glomeruli of MCD patient in relapse (A and D). Synaptopodin is expressed (red stain) in glomeruli of MCD patients in relapse (B). WT-1 is expressed (red stain) in glomeruli of MCD patients in relapse (E). CD80 and synaptopodin colocalized at the glomeruli (C). CD80 and WT-1 co-localized at the glomeruli (F).

Discussion

In the present study, we investigated the lymphocyte subsets, 34 cytokines expression of Th1/Th2/Th17 and the CD80 expression in adult-onset MCD. Our results showed that there was no significant difference in the proportion of CD3+CD4+ cells among patients with MCD in relapse, MCD in remission and healthy controls, but the cytokines GM-CSF and TNFSF11 were increased and the urinary CD80 levels were elevated in adult-onset MCD patients in relapse, indicating a disorder of Th1/Th2/Th17 balance and that the elevated excretion of CD80 may underlie the pathogenesis and development of adult-onset MCD.

Even though the proportion of CD3+CD4+ cells showed no significant difference between patients with MCD and healthy controls, the cytokines of GM-CSF and TRANCE increased significantly in patients with MCD in relapse, which might be due to the changes in CD3+CD4+ cell function. Interestingly, we found the proportions of CD8+ counts were elevated in MCD in relapse compared with controls, implicating CD8+ T cells might also participate in the course of MCD. GM-CSF, a representative cytokine of Th1 and Th17, is increased in the urine of patients with FSGS and is related to glomerulosclerosis (Stangou et al., 2017). Moreover, GM-CSF is secreted by renal parenchymal cells and inflammatory cells, which could mediate crescent formation, renal tubule injure and proteinuria, and ultimately lead to renal dysfunction in murine crescentic nephritis (Timoshanko et al., 2005). TRANCE, also known as RANKL, is a cytokine secreted by Th1 and its expression can be promoted by IL-17 which is a cytokine secreted by Th17 (Hienz, Paliwal & Ivanovski, 2015). However, the role of GM-CSF and TRANCE in patients with MCD is rarely reported. By analyzing the 34 cytokines secreted by Th1/Th2/Th17, we found that GM-CSF and TRANCE in the serum of patients with MCD in relapse were significantly higher than that in patients in remission. To our knowledge, we did not find the same results that there was increased expression of GM-CSF and TRANCE in the serum of patients with recurrent MCD. Accordingly, we speculate that the imbalance of Th1/Th2/Th17 in patients with MCD in relapse leads to the increase of GM-CSF and TRANCE secretion by Th1 and Th17 cells, which further induces podocyte damage and leads to proteinuria.

It has been proposed that CD80 plays a vital role in the “two-hit” podocyte immune disorder of MCD (Shimada et al., 2011). Our study showed that the excretion of urinary CD80 increased in adult-onset MCD patients in relapse when compared with MCD in remission and healthy controls. Similar results have been reported in other studies of childhood and adult-onset MCD patients (Garin et al., 2009; Zhao et al., 2018). However, we found there was no correlation between urinary CD80 and proteinuria in adult-onset MCD patients in relapse, demonstrating the elevated CD80 was not simply a reflection of proteinuria. Besides, the serum CD80 in patients with MCD in remission was not different from those in relapse, implicating that the elevated urinary levels could not be explained by higher serum concentrations. Synaptopodin is a highly expressed actin binding protein in podocyte (Yu et al., 2018). WT-1 is known to be expressed on podocyte in kidney (Funk et al., 2016). The double immunostaining for CD80 and synaptopodin, and CD80 and WT-1 in the glomerulus of MCD patient in relapse showed colocalization, confirming the source of the urinary CD80 in MCD patients in relapse was the podocyte. In general, CD80 cannot be expressed on podocytes, but in some glomerulopathies, its expression on the surface of podocyte is increased. One of the possible reasons is that podocytes are switched to an antigen presenting cell phenotype (Trimarchi, 2015). Although the staining in Fig. 4 was convincing, immunohistochemistry was only performed on a single biopsy and further studies are needed to verify these findings.

Moreover, CD80 may interact with CTLA-4, which plays a crucial role in cellular and humoral immunity (Greenwald, Freeman & Sharpe, 2005). We found that urinary or serum CTLA-4 levels were not significantly increased in MCD patients in relapse compared with those in remission. However, the urinary CD80 to CTLA-4 ratio was higher in MCD in relapse versus in remission, which could be due to the defective response of Treg from patients in relapse to produce CTLA-4. Garin et al. (2015) reported that abatacept was useful in the treatment of one patient with MCD, but not in FSGS. Thus, the CD80-CTLA-4 axis seems to play an important part in the mechanism of proteinuria in recurrent MCD. Drugs targeting the CD80-CTLA-4 axis may be expected to treat MCD in relapse in the future.

There are some limitations in our study. Firstly, the sample size is small, so more studies are needed to verify the expression of GM-CSF and TRANCE in patients with MCD in relapse. Secondly, further studies are needed to explore the mechanism of GM-CSF and TRANCE in podocyte injury of patients with MCD. Thirdly, the interaction between GM-CSF, TRANCE and CD80 pathway in podocyte injury needs to be elucidated.

Conclusions

In our study, the cytokines GM-CSF and TRANCE were increased in the serum and the urinary CD80 levels were elevated in adult-onset MCD, indicating a disorder of Th1/Th2/Th17 balance and that the elevated excretion of CD80 may underlie the pathogenesis and development of adult-onset MCD. Further studies are warranted to investigate the precise mechanism of the interaction between Th1/Th2/Th17 balance and CD80 during the course of MCD.

Supplemental Information

Supplemental Information 1 Raw data: CD80, CTLA-4, lymphocyte subsets and cytokine antibody microarray data

Click here for additional data file.

Additional Information and Declarations

Competing Interests

Author Contributions

Human Ethics

Microarray Data Deposition

Data Availability

The authors declare there are no competing interests.

Ping Chen conceived and designed the experiments, performed the experiments, analyzed the data, prepared figures and/or tables, authored or reviewed drafts of the paper, and approved the final draft.

Yan Chen conceived and designed the experiments, performed the experiments, analyzed the data, authored or reviewed drafts of the paper, and approved the final draft.

Maoqing Jiang conceived and designed the experiments, analyzed the data, prepared figures and/or tables, authored or reviewed drafts of the paper, and approved the final draft.

Yijun Mo and Huanhuan Ying performed the experiments, prepared figures and/or tables, and approved the final draft.

Xun Tang performed the experiments, authored or reviewed drafts of the paper, and approved the final draft.

Jun Zhang conceived and designed the experiments, analyzed the data, authored or reviewed drafts of the paper, and approved the final draft.

The following information was supplied relating to ethical approvals (i.e., approving body and any reference numbers):

Our study was approved by the Medical Ethics Committee of Ningbo First Hospital (Ethical Application Ref: 2017-R033).

The following information was supplied regarding the deposition of microarray data:

The 34 cytokines and microarray raw data are available as a Supplemental File.

The following information was supplied regarding data availability:

The raw measurements are available as Supplemental File.

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
