# Peer review of "Usefulness of the cytokines expression of Th1/Th2/Th17 and urinary CD80 excretion in adult-onset minimal change disease"

_PeerJ, doi:10.7717/peerj.9854_

## Round 0.1 · original submission · Major Revisions

Major concerns have to be addressed carefully by the authors.

Reviewer 1 ·

Basic reporting

No comment

Experimental design

No comment

Validity of the findings

No comment

Additional comments

Data are relatively novel, but of limited clinical impact. Whether this evaluation can be applied in clinical
practice, whether it can be handled by all labs and how expensive it is remain unaddressed issues.
Moreover, this paper suffers from some methodological limitations and results do not support any definite conclusion.
Major concerns:
- The number of pts is very limited
- More data should be provided about patient clinical history (proteinuria, previous treatment)

Reviewer 2 ·

Basic reporting

The article is written in professional English, technically correct and unambiguous.
The literature references are of sufficient support as background
Structure of the manuscript is professionaly desinged and the paper is self-contained

Experimental design

The primary research is wthin the aims and scope of the journal. The research question is well defined relvant and meaningful, and the whole study has a role to fill a knowledge gap
Investigation is performed to a high technical and ethical standard
Methods are described with details and give sufficient information to replicate.

Validity of the findings

The data are provided, results ar clear and novel, some negative results are discussed and explained. Conclusions are well stated and are limited to supporting results.

Additional comments

The paper is very well written, no comments for methods, results, discussion. Few typing errors, such as in line 160 word TRNACE must change to TRANCE

·

Basic reporting

The use of the English language requires major polishing. Too many grammatical mistakes. Needs to be corrected by a native English speaking person.
The references are adequate, despite some suggestiosn are given ahead.
Professional article structure is adequate.
There is coherence between hypothesis, findings are conclusions

Experimental design

The research has been undertaken appropriately.

Validity of the findings

This article offers new data to the literature.
The number of patients is low, a drawback that has been highlighted in the limitations of the study.
Conclusions are correctly addressed.

Additional comments

The present article written by Chen Ping et al is about some T cell sub-population cytokines in a small cohort of patients with minimal change disease (MCD), some in relapse and some in remission.
Concerns:
English language is poor, and requires major polishing. The article is sometimes hard to follow or interpret.
The number of cases is low to draw firm conclusions.
Abstract: Background, authors state that MCD is a common form of nephritic syndrome. This is completely wrong. MCD is a nephrotic entity, not nephritic, please.
Methods: THe age and gender of patients must be outlined in the abstract.
Abbreviatons must be explained what they stand for, as TRANCE, etc.
At the end of the Introduction, authors comment that "...few studies have reported the role of urinary CD80 excretion in adult-onset CD80..". This is wrong. Please, replace by: few studies have tried to explain the possible reasons why elevated urinary levels of CD80 excretion are encountered in adult-onset MCD".
How was MCD defined?. Was electron microscopy performed?. They cite Vivarelli et al 2017, a review paper based both for children and adults. The definition of MCD given by this review is broad and ample, and of course describes the findings in OM, IF and EM. So, authors must addess the eway MCD was defined: OM?. Was EM performed?.
Methods: Delete the words "on an empty stomach".
Auhtors must hypothesize why CD80 levels are elevated in the urine. The source appears to be podocytes. The why?. One of the possible reasons is that podocytes are switched to an antigen presenting cell phenotype. Please read and cite Trimarchi H: Recent Pat Endocr Metab Immune Drug Discov 2015;9(1):2-14. doi: 10.2174/1872214809666150302104542.

---

## Round 0.2 · accepted · Accept

The manuscript is now ready for publication.

Reviewer 1 ·

Basic reporting

No comment

Experimental design

No comment

Validity of the findings

No comment

Additional comments

The revision version of the paper partially improves its quality. However some criticalities persist. First of all the possibility of this evaluation in real life. Moreover, the number of pts remains very limited.
Taken together, these drawbacks make the manuscript unsuitable for publication.

Reviewer 2 ·

Basic reporting

no comment

Experimental design

no comment

Validity of the findings

no comment

Additional comments

no comment

·

Basic reporting

English language has been polished.
Besides, no changes as expressed previously in first submission

Experimental design

Correct

Validity of the findings

Correct

Additional comments

I have no additional comments